# Polymerization of misfolded Z alpha-1 antitrypsin protein lowers CX3CR1 expression in human PBMCs

Srinu Tumpara[1], Matthias Ballmaier[2], Sabine Wrenger[1], Mandy König[3], Matthias Lehmann[3], Ralf Lichtinghagen[4], Beatriz Martinez-Delgado[5], Elena Korenbaum[6], David DeLuca[1], Nils Jedicke[7], Tobias Welte[1], Malin Fromme[8], Pavel Strnad[8], Jan Stolk[9], Sabina Janciauskiene[1,9]*

[1]Department of Respiratory Medicine, Hannover Medical School, Biomedical Research in Endstage and Obstructive Lung Disease Hannover (BREATH), Member of the German Center for Lung Research (DZL), Hannover, Germany; [2]Cell Sorting Core Facility Hannover Medical School, Hannover, Germany; [3]8sens.biognostic GmbH, Berlin, Germany; [4]Institute of Clinical Chemistry, Hannover Medical School, Hannover, Germany; [5]Department of Molecular Genetics, Institute of Health Carlos III, Center for Biomedical Research in the Network of Rare Diseases (CIBERER), Majadahonda, Spain; [6]Institute for Biophysical Chemistry, Hannover Medical School, Hannover, Germany; [7]Department of Gastroenterology, Hepatology and Endocrinology, Hannover Medical School, Hannover, Germany; [8]Medical Clinic III, Gastroenterology, Metabolic Diseases and Intensive Care, University Hospital RWTH Aachen, Aachen, Germany; [9]Department of Pulmonology, Leiden University Medical Center, Member of European Reference Network LUNG, section Alpha-1-antitrypsin Deficiency, Leiden, Netherlands

**Abstract** Expression levels of CX3CR1 (C-X3-C motif chemokine receptor 1) on immune cells have significant importance in maintaining tissue homeostasis under physiological and pathological conditions. The factors implicated in the regulation of CX3CR1 and its specific ligand CX3CL1 (fractalkine) expression remain largely unknown. Recent studies provide evidence that host's misfolded proteins occurring in the forms of polymers or amyloid fibrils can regulate CX3CR1 expression. Herein, a novel example demonstrates that polymers of human ZZ alpha-1 antitrypsin (Z-AAT) protein, resulting from its conformational misfolding due to the Z (Glu342Lys) mutation in *SERPINA1* gene, strongly lower CX3CR1 mRNA expression in human peripheral blood mononuclear cells (PBMCs). This parallels with increase of intracellular levels of CX3CR1 and Z-AAT proteins. Presented data indicate the involvement of the CX3CR1 pathway in the Z-AAT-related disorders and further support the role of misfolded proteins in CX3CR1 regulation.

*For correspondence:
janciauskiene.sabina@mh-hannover.de

## Introduction

Interactions between the chemokine receptors and chemokines, but also other proteins, peptides, lipids, and microbial products, play a critical role in the recruitment of inflammatory cells into injured/diseased tissues (*Bachelerie et al., 2014*). Many human diseases involve altered surface expression of chemokine receptors, which can lead to a defective cell migration and inappropriate immune response. Most of the human peripheral blood mononuclear cells (PBMCs) express CX3CR1 (*Bazan et al., 1997*), also known as the G-protein-coupled receptor 13 (GPR13) or fractalkine receptor, a mediator of leukocyte migration and adhesion. In the central nervous system, CX3CR1 is

**eLife digest** Proteins can lose their structure and form polymers because of mutations or changes in their immediate environment which can lead to cell damage and disease. Interestingly, polymers formed by a variety of proteins can reduce the levels of CX3C chemokine receptor 1 (CX3CR1 for short) that controls the behaviour of immune cells and is implicated in a range of illnesses.

Inherited ZZ alpha-1 antitrypsin deficiency is a rare genetic condition that highly increases the risk of liver and lung diseases. This disorder is characterised by mutant alpha-1 antitrypsin proteins (AAT for short) reacting together to form polymers; yet it remains unclear how the polymers affect different cells or organs, and lead to diseases. To investigate this question, Tumpara et al. examined whether polymers of mutant AAT influence the level of the CX3CR1 protein in specific classes of immune cells.

Experiments revealed that in people with AAT deficiency, certain blood immune cells express lower levels of CX3CR1. Regardless of age, clinical diagnosis, or treatment regimen, all individuals with ZZ alpha-1 antitrypsin deficiency had AAT polymers circulating in their blood: the higher the levels of polymers measured, the lower the expression of CX3CR1 recorded in the specific immune cells. When Tumpara et al. added polymers of mutant AAT to the immune cells of healthy donors, the expression of CX3CR1 dropped in a manner dependent on the polymer concentration. According to microscopy data, AAT polymers occurred inside cells alongside the CX3CR1 protein, suggesting that the two molecular actors interact. In the future, new drugs that remove these polymers, either from inside cells or as they circulate in the body, could help patients suffering from conditions associated with this abnormal protein aggregation.

largely expressed by microglial cells (brain macrophages) (*Ransohoff, 2009*), which are involved in neurodegenerative diseases like Alzheimer's disease. The major role of CX3CR1-expressing cells is to recognize and enter tissue following CX3CL1 (fractalkine or also called neurotactin) gradient, and to crawl or 'patrol' in the lumen of blood vessels (*Auffray et al., 2007*). Since CX3CR1/CX3CL1 axis is also involved in the synthesis of anti-inflammatory cytokines and has a significant role in cytoskeletal rearrangement, migration, apoptosis, and proliferation, its dysregulation is associated with the development of cardiovascular diseases, kidney ischemia–reperfusion injury, cancer, chronic obstructive pulmonary disease (COPD), neurodegenerative disorders, and others (*Harrison et al., 1998*; *Ning et al., 2004*; *Rius et al., 2013*). Some studies indicate that CX3CR1 deficiency contributes to the severity of infectious diseases (*Bonduelle et al., 2012*) and promotes lung pathology in respiratory syncytial virus-infected mice (*Das et al., 2017*). Animals with deletion of CX3CR1 show impaired phagocytosis (*Thome et al., 2015*), which is vital to prevent unwanted inflammation. It is clear that CX3CR1-expressing cells have tissue-specific roles in different pathophysiological conditions. Nevertheless, a comprehensive knowledge on the regulation of CX3CR1 expression is still missing.

Current findings suggest that divergent proteins with a common propensity to form extracellular oligomers interact with chemokine receptors and affect their expression levels. For example, Alzheimer's peptide, Aβ, interacts with CX3CR1 and significantly reduces its expression in cultured microglial cells and in Alzheimer's brain (*Cho et al., 2011*). Similarly, highly aggregated extracellular Tau protein binds to CX3CR1, promotes its internalization, and reduces expression in microglial cells (*Bolós et al., 2017*). In concordance, polymers of human Z alpha-1 antitrypsin (Z-AAT), resulting from protein misfolding due to the Z (Glu342Lys) mutation in *SERPINA1* gene, lower CX3CR1 mRNA expression in human PBMCs, which parallels with increased intracellular CX3CR1 and Z-AAT protein levels.

## Results and discussion

Inherited alpha-1 antitrypsin deficiency (AATD) is a rare genetic condition caused by *SERPINA1* gene mutations. Homozygous Z AATD mutation is the most clinically relevant among Caucasians (prevalence is about 1:2000-1:5000) that is characterized by low plasma levels of AAT protein (10–15% compared to the wild type, MM AAT, 1.3–2 g/l) and the presence of intracellular and circulating Z-AAT polymers (*Tan et al., 2014*). The liver is the major producer of AAT, therefore the

accumulation of Z-AAT polymers in hepatocytes is a marker for diagnosing AATD (*Janciauskiene et al., 2011*). The intracellular Z-AAT polymers have also been identified in other AAT-expressing cells like monocytes and macrophages (*Belchamber et al., 2020*). The accumulation of polymers is harmful for AAT-producing cells, whereas the circulating Z-AAT polymers are not able to execute the tasks of AAT protein, a major inhibitor of serine proteases having a strong immuno-modulatory potential. Based on the facts that: (i) circulating Z-AAT polymers contribute to the risk of developing pathologies (*Parmar et al., 2002*; *Strnad et al., 2020*), (ii) pathogenic oligomeric proteins affect CX3CR1 expression (*Bolós et al., 2017*), and (iii) CX3CR1/CX3CL1 axis plays a significant role in immunity (*Imai and Yasuda, 2016*), we aimed to investigate CX3CR1 expression in PBMCs of ZZ AATD individuals. For this, in collaboration with German Alpha1 Patient Association and Aachen University, was prepared RNA from freshly isolated PBMCs of 41 clinically stable ZZ AATD volunteers independently of their clinical diagnosis or treatment with intravenous AAT, a specific augmentation therapy (*Janciauskiene and Welte, 2016*). For comparison, PBMCs isolated from healthy volunteers having normal plasma AAT levels were used. Additionally, a limited amount of RNA sample was available from PBMCs isolated from a cohort of 12 ZZ AATD emphysema patients at Leiden University Medical Center, The Netherlands (*Figure 1—figure supplement 1*).

Independent of individual's age, clinical diagnosis (healthy, lung or liver disease), or augmentation therapy, the *CX3CR1* mRNA expression turned to be much lower in ZZ AATD PBMCs than in PBMCs from non-AATD controls (median [range]: 4.1 [2.7–5.5] vs. 18.5 [13–26.6], p<0.001) (*Figure 1A*). A previous study has shown that CX3CR1$^{-/-}$ mice have significantly higher plasma levels of CX3CL1 than wild-type mice (*Cardona et al., 2008*). A diminished expression of CX3CR1 might be related to increased levels of soluble CX3CL1, an exclusive ligand for CX3CR1 (*Bachelerie et al., 2014*). However, the concentration of plasma CX3CL1 was low, did not differ between ZZ AATD and non-AATD individuals (*Figure 1B*), and did not correlate with *CX3CR1* mRNA in PBMCs. The expression and release of CX3CL1 is generally low in the absence of inflammatory insults (*Umehara et al., 2004*) showing that at the time point of blood donation all volunteers were under stable clinical condition.

Although CX3CR1 is preferentially expressed on monocytes, other cells also express this receptor (*Landsman et al., 2009*). Previous reports indicated that exogenous IL-15 is a negative regulator of

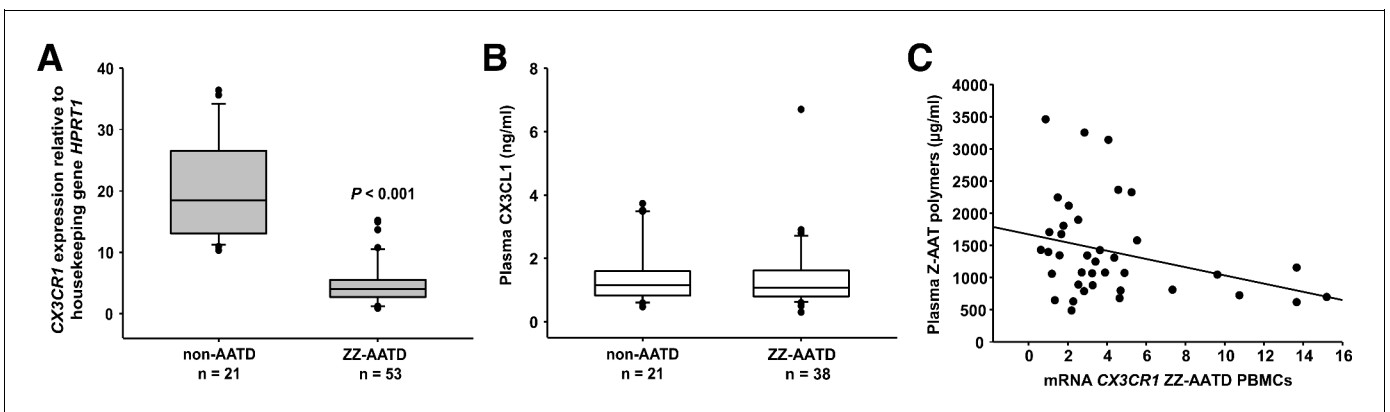

**Figure 1.** Relationship between CX3CR1 gene expression levels and AAT deficiency and the concentration of circulating AAT polymers. *CX3CR1* gene expression levels in peripheral blood mononuclear cells (PBMCs) related to alpha-1 antitrypsin-deficiency (AATD) and plasma concentrations of CX3CR1 ligand (CX3CL1) and Z-AAT polymer. (**A**)PBMCs were isolated from AATD subjects and non-AATD controls. *CX3CR1* gene expression relative to *HPRT1* housekeeping gene was determined by real-time PCR using Taqman gene expression assays. Measurements were carried out in duplicates. Data are presented as median (IQR) in boxplots, lines represent medians. Outliers are defined as data points located outside the whiskers. p-Value was calculated by Mann-Whitney U test. (**B**) Plasma levels of CX3CL1 in AATD (plasma available for n = 38 AATD) and non-AATD individuals measured by ELISA. Measurements were carried out in triplicates. Data are presented as median (IQR) in boxplots with whiskers. Outliers are defined as data points located outside the whiskers. (**C**) Negative correlation of *CX3CR1* mRNA in PBMCs and plasma Z-AAT polymer levels from ZZ AATD individuals from graph (**B**). Pearson's correlation test, $r^2$ = −0.313, p=0.055, n = 38.

The online version of this article includes the following source data and figure supplement(s) for figure 1:

**Source data 1.** Source files, containing original data for *Figure 1A and B*, to document *CX3CR1* expression (**A**), and plasma levels of CX3CL1 in alpha-1 antitrypsin-deficient (AATD) and non-AATD individuals (**B**).

**Figure supplement 1.** Schematic presentation of the study design.

CX3CR1 expression in human CD56[+] NK cells (*Barlic et al., 2003*; *Sechler et al., 2004*). However, plasma levels of IL-15 were lower in ZZ AATD than in non-AATD (pg/ml, median [range]: 6.6 [5.9–6.9], n = 23 vs. 7.63 [6.63–8.1], n = 21, p=0.001), excluding a possible link between IL-15 and *CX3CR1* mRNA levels.

Because ZZ AATD individuals, differently from non-AATD, have about 90% lower blood concentration of Z-AAT protein, which may influence cellular microenvironment (*Ramos et al., 2010*), a relationship between *CX3CR1* and Z-AAT plasma levels cannot be excluded. However, no correlation was found between *CX3CR1* mRNA in PBMCs and plasma levels of Z-AAT measured by nephelometry (data not shown). Next, plasma Z-AAT polymers were measured, as the biomarkers of all carriers of the Z allele (*Tan et al., 2014*; *Janciauskiene et al., 2002*). As anticipated, only minor amounts of polymers were detected in plasma of non-AATD individuals while plasma of ZZ AATD contained high amounts of polymers (µg/ml, mean [SD]: 4.1 [6], n = 18 vs. 1399.8 [750], n = 20, respectively). Since most of the ZZ AATD individuals received intravenous augmentation therapy with plasma purified AAT protein, ZZ AATD individuals were segregated into subgroups who receive or not receive therapy. There were no significant differences in Z-AAT polymer levels between the subgroups: (µg/ml, median [range]: non-augmented 1506.6 [854–1781], n = 17 vs. augmented 1348.5 [779.5–1529], n = 23, respectively). A previous study used a sandwich ELISA based on 2C1 antibody and found that circulating Z-AAT polymers range between 8.2 and 230.2 µg/ml in ZZ AATD (*Tan et al., 2014*) whereas much higher circulating levels of Z-AAT polymers were detected by using the single monoclonal antibody (LG96)-based ELISA. These discrepancies can be due to the differences between antibody specificities. For example, 2C1 showed high affinity for polymers formed by heating M- or Z-AAT at 60˚C (*Miranda et al., 2010*) while LG96 antibody recognizes naturally occurring/native Z-AAT polymers without requiring sample heating. To answer, why some individuals have higher plasma levels of Z-AAT polymers than monomers (measured by nephelometry) is of great importance for the further studies.

Most interestingly, in ZZ AATD individuals, was found a trend toward an inverse relationship between *CX3CR1* mRNA in PBMCS and plasma Z-AAT polymers ($r^2 = -0.31$, n = 38, p=0.055) (*Figure 1C*). This latter finding prompted more extensive investigation whether Z-AAT polymers affect CX3CR1 expression when added to healthy donor PBMCs for 18 hr, ex vivo. Lipopolysaccharide (LPS, from *Escherichia coli*, 1 µg/ml) was included as a known reducer of CX3CR1 expression (*Pachot et al., 2008*; *Sica et al., 1997*). Indeed, polymeric Z-AAT in a concentration-dependent manner lowered *CX3CR1* mRNA (*Figure 2—figure supplement 2*) whereas repeated experiments using Z-AAT at a constant concentration of 0.5 mg/ml reduced *CX3CR1* mRNA more than twice as compared to non-treated controls (*Figure 2A*). Accordingly, LPS and polymer containing Z-AAT preparation significantly decreased surface expression of CX3CR1, specifically in CD14[+] monocytes and NK cells (*Figure 3*).

By contrast, cellular levels of CX3CR1 protein increased in PBMCs treated with Z-AAT polymers or LPS (used as a positive control) as compared to non-treated controls (*Figure 2B*). The CX3CR1 protein was present in detergent resistant lipid raft fraction of PBMCs treated with Z-AAT (*Figure 2—figure supplement 3A*). Total cell lysates and lipid raft fractions from Z-AAT-treated PBMCs, in contrast to those prepared from M-AAT-treated or non-treated PBMCs, contained high amounts of AAT polymers (*Figure 2C*, *Figure 2—figure supplement 3B*). The laser scanning confocal microscopy of double-labeled specimens showed a co-localization of Z-AAT polymers with CX3CR1 protein (*Figure 2D*). Furthermore, 3D reconstruction of cross sections visualized larger Z-AAT aggregates surrounded by cellular extensions in a cap-like formation, suggesting that cells may react differently depending on the size of Z-AAT polymers (*Figure 2E*). It cannot be excluded that Z-AAT polymers, similar like polymers of Tau protein, interact with CX3CR1 and get internalized (*Chidambaram et al., 2020*). This may determine the fate of CX3CR1 mRNA expression, that is, sequestered intracellularly and not returning to the cell surface, CX3CR1 protein may induce signaling pathways lowering *CX3CR1* expression. To achieve a definitive answer how Z-AAT or other types of protein polymers regulate CX3CR1 levels, detailed mechanistic studies are required. In general, along with transcriptional regulation, chemokine receptors trafficking is of great importance to understand (*Kershaw et al., 2009*).

Under the same experimental conditions, monomeric M-AAT had no effect on CX3CR1 mRNA expression relative to housekeeping gene HPRT1 (mean [SD]: 24.9 [2.9] controls, n = 5 vs. 23.7 [1.3], n = 5, NS), and protein levels (*Figure 2B*). Likewise, monomeric Z-AAT protein does not affect

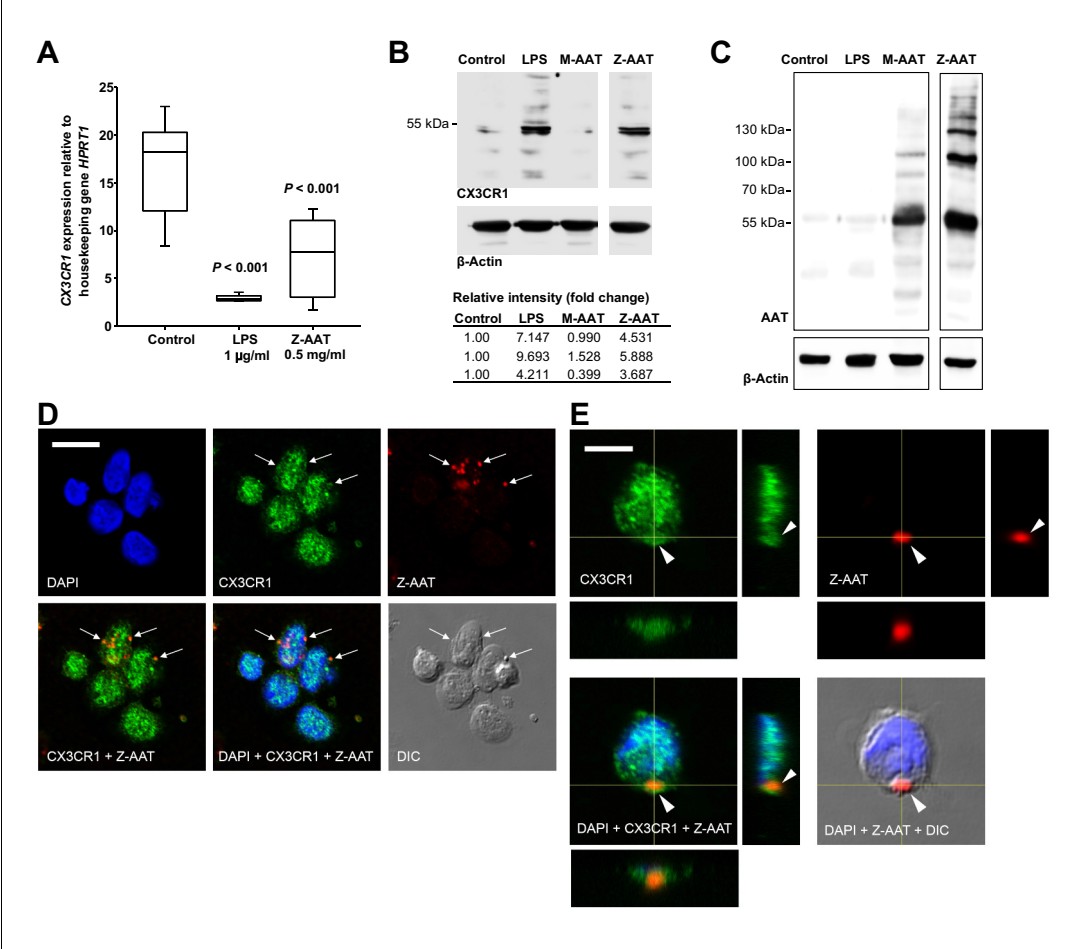

**Figure 2.** Effects of Z alpha-1 antitrypsin (Z-AAT) and M-AAT on CX3CR1 mRNA and protein expression. (**A**) *CX3CR1* gene expression relative to *HPRT1* housekeeping gene was determined by real-time PCR using Taqman gene expression assays. Peripheral blood mononuclear cells (PBMCs) were incubated for 18 hr with plasma-derived Z-AAT, lipopolysaccharide (LPS), or M-AAT in the concentrations as indicated, or with RPMI medium alone (control). The data from n = 6 independent experiments are presented as median (IQR) in box and whisker plot format; lines represent medians in each box. Measurements were carried out in duplicates. p-Value was calculated by nonparametric Kruskal-Wallis test. (**B**) Representative uncut Western blot (n = 3 independent experiments) of CX3CR1 in RIPA lysates prepared from PBMCs incubated for 18 hr alone or with LPS (1 μg/ml), M-AAT (1 mg/ml), or Z-AAT (0.5 mg/ml). For analysis of CX3CR1, equal amounts of protein were separated by SDS-PAGE under reducing conditions. Relative intensities were calculated for each band using the ratio relative to β-actin, as a loading control, and then normalized by the experimental control. (**C**) For analysis of cellular AAT, the same lysates were separated under non-reducing conditions. Western blots were probed with polyclonal rabbit anti-human AAT recognizing monomeric, polymeric, or truncated forms of AAT. One representative blot from n = 3 independent experiments is shown. β-Actin was used for a loading control. (**D**) and (**E**) Co-distribution of Z-AAT polymers with CX3CR1 in human total PBMCs incubated with 0.5 mg/ml Z-AAT polymers for 18 hr. (**D**) Immunofluorescence microscopy revealed co-localization of Z-AAT polymers (*red*) with CX3CR1-positive structures (*green*). Arrows point areas of co-localization. Scale bar, 10 μm. (**E**) Confocal microscopy 3D stack with orthogonal reconstruction shows an aggregate of Z-AAT polymers (*red*) surrounded by CX3CR1-positive (*green*) cellular extensions forming a cap-like structure (arrowhead). Scale bar, 5 μm. The images with indicated channels merged and the corresponding differential interference contrast (DIC) image are presented. 4', 6- Diamidino 2-phenylindole (DAPI) was used for nuclei staining (*blue*).

The online version of this article includes the following source data and figure supplement(s) for figure 2:

**Source data 1.** Source file, containing original data for *Figure 2A*, to document reduced *CX3CR1* expression in peripheral blood mononuclear cells (PBMCs) treated with Z alpha-1 antitrypsin (Z-AAT) or lipopolysaccharide (LPS) (**A**).

**Figure supplement 1.** Quality control of isolated M- and Z-AAT (alpha-1 antitrypsin) proteins by SDS-PAGE.

**Figure supplement 2.** Z alpha-1 antitrypsin (Z-AAT) in a concentration-dependent manner reduces *CX3CR1* mRNA expression in peripheral blood mononuclear cells (PBMCs) isolated from healthy donors.

**Figure supplement 3.** Z alpha-1 antitrypsin (Z-AAT) induces association of CX3CR1 with lipid rafts.

**Figure supplement 4.** CX3CR1 transcript and protein expressions in the presence of Z alpha-1 antitrypsin (Z-AAT) monomer, Z-AAT polymer, native M-AAT, and M-AAT polymer.

**Figure supplement 5.** Effect of CX3CL1 alone or in combination with Z alpha-1 antitrypsin (Z-AAT) on CX3CR1 transcript and protein expression.

*Figure 2 continued on next page*

*Figure 2 continued*

**Figure supplement 5—source data 1.** Source file, containing original data for *Figure 2—figure supplement 5A*, to document *CX3CR1* gene expression in peripheral blood mononuclear cell (PBMC).

CX3CR1 mRNA and protein levels, and heat-induced polymers of M-AAT showed no effect on CX3CR1 expression as well (*Figure 2—figure supplement 4*). Probably, specific conformational properties and/or molecular sizes of Z-AAT polymers are required for their interaction with CX3CR1. For example, cell surfaces express CX3CL1 as a constitutive oligomer (three to seven molecules), which is essential for efficient interaction with CX3CR1 (*Hermand et al., 2008*; *Ostuni et al., 2020*). Numerous chemokines tend to self-associate that determines their activity (*Proudfoot et al., 2003*), and therefore certain Z-AAT polymers may resemble chemokine structures competing for the same receptor(s). In some experimental models, Z-AAT polymers expressed strong chemotactic properties (*Parmar et al., 2002*; *Lomas and Mahadeva, 2002*). When chemokine receptors are engaged in chemotaxis, they can be removed from the cell surface by the ligand–receptor internalization (*Springer, 1994*), which might explain a decrease of CX3CR1 in ZZ PBMCs. Interestingly, the soluble form of CX3CL1, even when used at a high concentration of 500 ng/ml, does not antagonize Z-AAT polymer effects on CX3CR1 mRNA and protein levels, and by itself showed no effect on CX3CR1

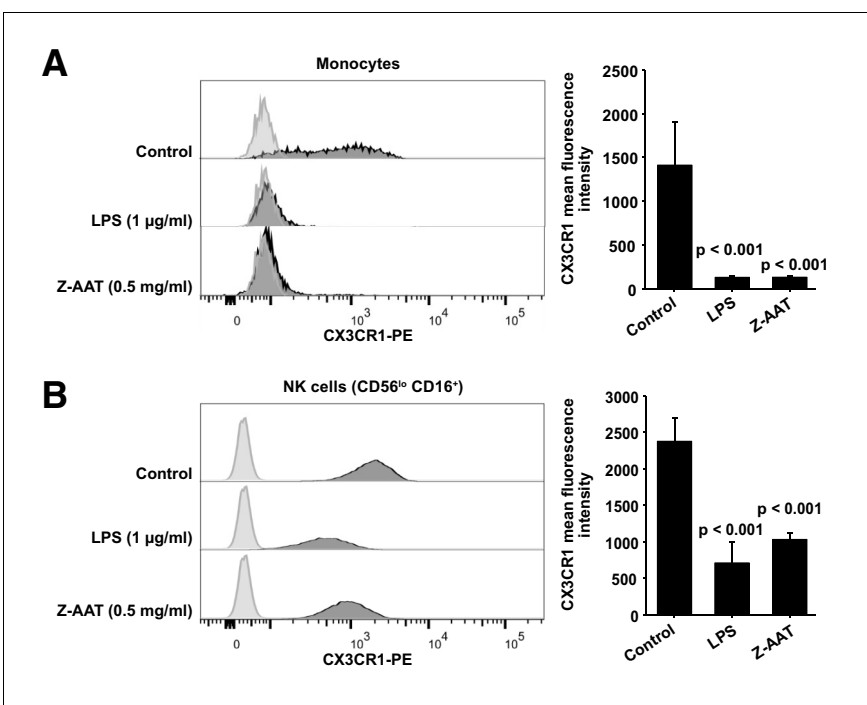

**Figure 3.** Flow cytometric analysis of CX3CR1 surface expression in peripheral blood mononuclear cells (PBMCs) after incubation with RPMI alone (control), Z alpha-1 antitrypsin (Z-AAT), or lipopolysaccharide (LPS) in the concentrations as indicated for 18 hr. CX3CR1-expressing cells were found in the monocyte gate (**A**) and in the NK cell gate (CD56[lo] CD16[+]) (**B**). Histograms show representative results and bars represent mean (SD) of n = 4 independent biological repeats each measured one time. After incubation with Z-AAT or LPS, monocytes and NK cells show significantly reduced CX3CR1 surface expression in comparison to untreated control cells. p-Values were calculated by one-way ANOVA.

The online version of this article includes the following source data and figure supplement(s) for figure 3:

**Source data 1.** Source files, containing original data for *Figure 3A,B*, to document reduced CX3CR1 surface expression in monocytes (**A**) and NK cells (**B**) after treatment with Z alpha-1 antitrypsin (Z-AAT) or lipopolysaccharide (LPS) (**A**).

**Figure supplement 1.** Gating strategy: sequential gating to identify monocytes and NK cells from total peripheral blood mononuclear cells (PBMCs).

mRNA or protein levels (*Figure 2—figure supplement 5*), although some studies reported that CX3CL1 reduces CX3CR1 expression (*Pachot et al., 2008*; *White et al., 2014*). In solution CX3CL1 remains monomeric, even at high concentrations (*Hermand et al., 2008*; *Mizoue et al., 1999*) whereas, as mentioned above, membrane-bound CX3CL1 occurs as oligomer. These two forms of the CX3CL1 perform differential roles (*Winter et al., 2020*), and therefore it cannot be excluded that oligomeric, but not a soluble, form of CX3CL1 would compete with Z-AAT for CX3CR1 interaction in vivo.

The CX3CR1 helps to define the major subsets of human monocytes because classical monocytes express much lower levels of CX3CR1 than non-classical monocytes (*Ziegler-Heitbrock et al., 2010*). After in vitro challenge with LPS for longer periods (like for 18 hr), human monocytes are known to increase in the mRNA and membrane expression of CD14, a receptor for LPS (*Landmann et al., 1996*). The enhancement of CD14 expression after treatment of PBMCs with Z-AAT strikingly resembled LPS (*Figure 4* and *Figure 4—figure supplement 1*). This raised a suspicion that Z-AAT preparations might contain endotoxin. According to the limulus amebocyte lysate test, endotoxin levels of Z-AAT preparations were below detection limit (0.01 EU/ml). Moreover, LPS significantly induced expression of TNFα, IL-6, and IL-1β while polymer containing Z-AAT preparations had no effect (*Figure 4—figure supplement 2*). Besides, LPS but not Z-AAT significantly increased release of cytokines (IL-1β, pg/ml, median [range]: LPS 1342.9 [1008–1834] vs. Z-AAT 3.2 [2.5–5.9] vs. controls 2.5 [2.1–3.7], n = 4 independent experiments; TNFα, ng/ml, mean [SD]: LPS 19.5 [2.5] vs. Z-AAT vs. controls, undetectable, n = 4 and IL-6, ng/ml, median [range]: LPS 15903.5 [14,626–17,262] vs. Z-AAT 5.4 [2.9–6.1] vs. control [1.7 (1.0–2.3), n = 4]). Therefore, the effect of Z-AAT preparations on CD14 is valid and unrelated to a potential LPS contamination. Although both Z-AAT polymers and LPS induce CD14 expression and similarly affect CX3CR1 expression and protein levels, data imply that Z-AAT polymers and LPS do not share the same signaling mechanisms.

As a side note, it has been reported that CD14$^{++}$ monocytes have the lowest expression of CX3CR1 (*Appleby et al., 2013*). Low and high surface CX3CR1 levels are suggested to delineate

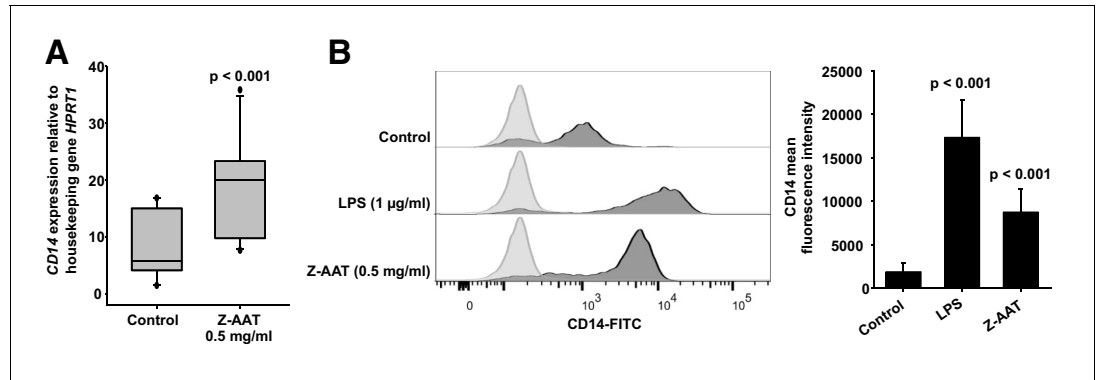

**Figure 4.** Z alpha-1 antitrypsin (Z-AAT) and lipopolysaccharide (LPS) induce CD14 expression. (**A**) Z-AAT increases *CD14* gene expression. Peripheral blood mononuclear cells (PBMCs) were incubated for 18 hr with 0.5 mg/ml Z-AAT or RPMI medium alone (control). *CD14* mRNA expression relative to HPRT1 was determined by real-time PCR using Taqman gene expression assays. Measurements were carried out in duplicates. Data are represented as median (IQR) in boxplots, lines represent medians of n = 14 independent biological repeats. Outliers are defined as data points located outside the whiskers. p-Values were calculated using nonparametric Kruskal-Wallis test. (**B**) Z-AAT increases monocyte CD14 surface expression. PBMCs were cultured with RPMI (control), Z-AAT, or LPS for 18 hr. CD14 mean fluorescence intensities of monocytic cells were determined by flow cytometry. Histograms show representative results and bars represent mean (SD) of n = 4 independent biological repeats each measured one time. p-Values were calculated from ANOVA.

The online version of this article includes the following source data and figure supplement(s) for figure 4:

**Source data 1.** Source files, containing original data for *Figure 4A, B* to document *CD14* gene expression in peripheral blood mononuclear cells (PBMCs) (**A**) and CD14 surface expression in monocytes (**B**).

**Figure supplement 1.** Inverse changes in *CD14* and *CX3CR1* mRNA expression in peripheral blood mononuclear cells (PBMCs) treated with different concentrations of Z alpha-1 antitrypsin (Z-AAT).

**Figure supplement 2.** Z alpha-1 antitrypsin (Z-AAT) does not induce cytokine expression.

**Figure supplement 2—source data 1.** Source files, containing original data for *Figure 4—figure supplement 2* to document *IL1B* (a), *IL6* (b), and *TNFA* (c) gene expression in peripheral blood mononuclear cells (PBMCs).

two functional subsets of murine blood monocytes: 'inflammatory' and 'resident monocytes' (*Geissmann et al., 2003*). This dichotomy appears conserved in humans as CD14$^+$ CD16$^-$, and CD14$^{low}$ CD$^{16+}$ monocytes resemble 'inflammatory' and 'resident' monocytes. Previous study demonstrated that peripheral blood monocytes of clinically healthy young adults (30 years of age) with ZZ AATD have significantly higher mRNA and surface expression of CD14 as compared to age matched MM subjects (*Sandström et al., 2008*). Authors thought that the higher CD14 expression reflects early pathological processes whereas according to the current findings this phenomenon seems to relate with the circulating Z-AAT polymers.

During steady state, non-classical monocytes expressing CX3CR1 patrol healthy tissues through crawling on the resting endothelium but these monocytes are required for a rapid tissue invasion at the site of infection or inflammation (*Auffray et al., 2009*; *Cros et al., 2010*). Previous work evidenced that the non-classical subset of monocytes, characterized by high expression of CX3CR1, is almost absent in ZZ AATD emphysema patients (*Stolk et al., 2019*). Moreover, plasma levels of AAT polymers were found to correlate with the levels of endothelium-related markers like sE-selectin and sICAM-1 (*Aldonyte et al., 2004*). Beyond, in a small cohort of ZZ AATD emphysema patients, was found a strong inverse association between lung function, based on percentage (%) predicted transfer factor for carbon monoxide (TLCO%pred) and forced expiratory volume in 1 s (FEV1%pred), and plasma levels of Z-AAT polymers: (TLCO%pred [$r^2 = -0.75$, n = 9, p=0.02] and FEV1%pred [$r^2 = -0.82$, n = 9, p=0.006]). Thus, higher levels of Z-AAT polymers and lower numbers of CX3CR1-positive cells may favor the development of lung injury and disease. A decrease in the expression of CX3CR1 on human monocytes has been shown in patients with atopic dermatitis (*Echigo et al., 2004*) and septic shock (*Pachot et al., 2008*).

To date, many functional aspects of the CX3CR1–CX3CL1 axis have been suggested, including the adhesion of immune cells to vascular endothelial cells, chemotaxis, the crawling of the monocytes that patrol on vascular endothelial cells, the retention of monocytes of the inflamed endothelium to recruit inflammatory cells, and the survival of the macrophage. Considering the above, these different aspects of interactions between PBMCs and Z-AAT or other polymers occurring due to genetic or post-translational protein modifications require further investigations in dedicated clinical and experimental studies.

# Materials and methods

## Key resources table

| Reagent type (species) or resource | Designation | Source or reference | Identifiers | Additional information |
|---|---|---|---|---|
| Biological sample (*Homo sapiens*) | PBMCs | Blood samples | | Collected from 41 ZZ AATD and 21 non-AATD healthy controls |
| Sequenced-based reagent (human) | CX3CR1 | Thermo Fisher Scientific | Taqman assay 4331182 | Hs00365842_m1 |
| Sequenced-based reagent (human) | TNFA | Thermo Fisher Scientific | Taqman assay 4331182 | Hs01113624_g1 |
| Sequenced-based reagent (human) | IL6 | Thermo Fisher Scientific | Taqman assay 4331182 | Hs00985639_m1 |
| Sequenced-based reagent (human) | IL1B | Thermo Fisher Scientific | Taqman assay 4331182 | Hs01555410_m1 |
| Sequenced-based reagent (human) | HPRT1 | Thermo Fisher Scientific | Taqman assay 4331182 | Hs02800695_m1 |

*Continued on next page*

*Continued*

| Reagent type (species) or resource | Designation | Source or reference | Identifiers | Additional information |
|---|---|---|---|---|
| Sequenced-based reagent (human) | CD14 | Thermo Fisher Scientific | Taqman assay 4331182 | Hs00169122_g1 |
| Other | Non-adherent 12-well plates | Greiner Bio-One | 665970 | |
| Other | Alpha-1 Antitrypsin Select matrix | GE Healthcare Life Sciences, Cytiva | 17547201 | |
| Other | Plasma purified human AAT | CSL Behring | Respreeza | |
| Other | LPS | Sigma-Aldrich | L2880 | *Escherichia coli* O55:B5 |
| Commercial assay or kit | Pierce Chromogenic Endotoxin Quant Kit | Thermo Fisher Scientific | A39553 | |
| Commercial assay or kit | UltraRIPA kit | BioDynamics Laboratory | F015 | |
| Antibody | Anti-AAT polymer, LG96 (mouse monoclonal) | Deposited at German Collection of Microorganisms and Cell Cultures: DSM ACC3092 | LG96 | ELISA: Coating: 1 µg/ml; LG96-HRP conjugate: (1:2000) Immunofluorescence: (1:5000) |
| Antibody | Anti-human AAT (rabbit polyclonal) | Agilent Dako | A001202-2 | (1:800) |
| Antibody | Anti-AAT polymer (mouse monoclonal) | Hycult Biotech | Clone 2C1; HM2289 | (1:500) |
| Antibody | Anti-CX3CR1 (rabbit polyclonal) | Abcam | ab8021 | (1:500) |
| Antibody | Anti-β-actin (mouse monoclonal) | Sigma-Aldrich | Clone AC-15; A3854 | HRP-conjugated; (1:20,000) |
| Antibody | Anti-CX3CR1 (rabbit polyclonal) | Abcam | ab8021 | (1:500) |
| Antibody | Anti-CX3CR1 (mouse monoclonal) | Invitrogen, Thermo Fisher Scientific | Clone 2A9-1; 12-6099-42 | PE-conjugated; 5 µl per test |
| Antibody | Anti-CD14 (mouse monoclonal) | Invitrogen, Thermo Fisher Scientific | Clone TuK4; MHCD1401 | FITC-conjugated; 5 µl per test |
| Antibody | Anti-CD16 (mouse monoclonal) | Immunotools | Clone 3G8; 21278166 | APC-conjugated; 5 µl per test |
| Antibody | Anti-CD56 (mouse monoclonal) | BD Biosciences | Clone NCAM16.2; 566124 | BV-480 conjugated; 5 µl per test |
| Peptide, recombinant protein | Human CX3CL1/Fractalkine | Bio-Techne | 365-FR-025 | |
| Software, algorithm | FlowJo | Becton, Dickinson and Company | | v10 |
| Software, algorithm | SigmaPlot 14 | Systat Software | | v14.0 |

## Study approval

The study cohort consists of 41 clinically stable ZZ AATD volunteers collected in collaboration with German Alpha1 Patient Association and Aachen University independently on their clinical diagnosis or treatment with intravenous AAT and 21 non-AATD healthy controls. The institutional review board of Aachen University (EK 173/15) provided ethical approval for individuals recruited in Germany. For Z-AAT polymer determination, we added 12 ZZ AATD emphysema patients recruited at Leiden University Medical Center. In addition, ZZ AATD emphysema patients (four males and five females) were enrolled with mean (SD): age 51 (6.6) years, forced expiratory volume in 1 s percent predicted (FEV1%pred, 66.3; *Pachot et al., 2008*) and transfer factor of the lung for carbon monoxide percent predicted (TLCO%pred, 64; *Sica et al., 1997*). The plasma levels of Z-AAT polymers in these cases were median (range) 714.2 ([412.9–2270.4] µg/ml). Leiden University Medical Center provided ethical approval (project P00.083 and P01.101) for the additional study groups. For all individuals, detailed medical records data were anonymized. All participants issued a written informed consent according

to the ethical guidelines of the Helsinki Declaration (Hong Kong Amendment) as well as Good Clinical Practice (European guidelines).

## Isolation of PBMCs

Total PBMCs were isolated from freshly obtained peripheral blood (within 6 hr) using Lymphosep (C-C-Pro, Oberdorla, Germany) discontinuous gradient centrifugation according to the manufacturer's instructions as described previously (*Frenzel et al., 2014*). Thereafter, cells were lysed with RLT buffer for RNA analysis or suspended in RPMI-1640 medium (Gibco, Thermo Fisher Scientific, Waltham, MA) and plated into non-adherent 12-well plates (Greiner Bio-One, Kremsmünster, Austria) for the further analyses.

## Real-time PCR analysis

Isolation of total RNA, synthesis of cDNA and mRNA analysis using Taqman gene expression assays (Thermo Fisher Scientific, Waltham, MA, *Table 1*) were performed as described previously (*Frenzel et al., 2014*). Real-time PCR was carried out in duplicates. RNA quality was checked on agarose gels.

## AAT polymer ELISA

The AAT polymer ELISA using the monoclonal antibody LG96 (deposited under access number DSM ACC3092 at German Collection of Microorganisms and Cell Cultures) was developed by Candor Biosciences. Normal M-AAT was used for a negative control. Recovery ratio, signal-to-noise ratio, calibration curve, sample stability under different storage conditions were tested and all tests passed. A cross-reactivity with M-AAT was not reported in any of the tests. Nunc MaxiSorp flat-bottom 96-well plates (Thermo Fisher, Waltham, MA) were coated overnight at 2–8°C with monoclonal antibody LG96, at 1 µg/ml in coating buffer pH 7.4 (Candor Biosciences, Wangen, Germany). After a 2 hr blocking step, the plasma samples were applied in the previously determined dilutions made in Low-Cross-Buffer (Candor Biosciences), which also served as a blank. Incubation was performed for 2 hr at room temperature (RT). For detection, the captured antigen was incubated with antibody (LG96)-horseradish peroxidase (HRP) conjugate (1:2000) for 2 hr. The conjugate was prepared in advance with the HRP Conjugation Kit Lightning-Link (Abcam, Cambridge, UK) according to the manufacturer's instructions. For signal development SeramunBlau fast2 microwell peroxidase substrate (Seramun, Heidesee, Germany) was used. The incubation was performed at RT for 12 min in the dark and the reaction was stopped with 2 M $H_2SO_4$. Plates were analyzed at 450 nm by microplate reader (Dynex, Chantilly, VA) equipped with Dynex Revelation 4.21 software. Measurements were carried out in triplicates.

## Preparation of AAT proteins

Plasma M- and Z-AAT was isolated by affinity chromatography using the AAT-specific Alpha-1 Antitrypsin Select matrix (GE Healthcare Life Sciences, Cytiva, Sheffield, UK) according to the manufacturer's recommendations. For Z-AAT preparation plasma from volunteers not receiving AAT augmentation therapy was pooled. To change the buffer in the M- and Z-AAT protein pools to Hank's balanced salt solution (HBSS, Merck, Darmstadt, Germany), Vivaspin centrifugal concentrators with 10,000 MWCO (Vivaproducts, Littleton, MA) were used. Plasma purified human AAT (99% purity, Respreeza, Zemaira, CSL Behring, Marburg, Germany) was changed to HBSS by the same

**Table 1.** Primers for gene expression analysis.

| Primer | Assay ID |
| --- | --- |
| CX3CR1 | Hs00365842_m1 |
| TNFA | Hs01113624_m1 |
| IL6 | Hs00985639_m1 |
| IL1B | Hs01555410_s1 |
| HPRT1 | Hs02800695_m1 |
| CD14 | Hs00169122_g1 |

method. Protein concentrations were determined using Pierce BCA Protein Assay Kit (Thermo Fisher, Waltham, MA). The quality of the M- and Z-AAT preparations was confirmed on Coomassie gels (10% SDS-PAGE, *Figure 2—figure supplement 1A*) and by analyzing endotoxin levels with Pierce Chromogenic Endotoxin Quant Kit according to the manufacturer's guidelines (Thermo Fisher, Waltham, MA) using TECAN Infinite M200 PRO (Männedorf, Switzerland). In both, M- and Z-AAT preparations, endotoxin levels were below the detection limit (assay sensitivity: 0.01–0.1 EU/ml).

## Preparation of Z-AAT monomers

Z-AAT was isolated by affinity chromatography using AAT-specific Alpha-1 Antitrypsin Select matrix as described above. After the isolation Z-AAT, protein was diluted with sterile 0.9% NaCl (Fresenius Kabi, Bad Homburg, Germany), and Vivaspin-20, 100 kDa centrifugal column units (Sartorius, Göttingen, Germany) were used to separate Z-AAT monomers from polymers. Protein concentrations were determined using the Pierce BCA Protein Assay Kit (Thermo Fisher Scientific, Carlsbad, CA) according to manufacturer's instructions. The Z-AAT protein monomers were confirmed by using 7.5% SDS-PAGE without sample heating and without β-mercaptoethanol (*Figure 2—figure supplement 1B*).

## In vitro experiments with PBMCs from healthy donors

PBMCs ($5 \times 10^6$ cells/ml) were incubated for 18 hr at 37°C, 5% $CO_2$ either alone, or with Z- or M-AAT proteins, or LPS (1 μg/ml, *E. coli O55:B5*, Sigma-Aldrich, Merck, St. Louis, MO). In some experiments, a recombinant CX3CL1 protein (R&D Systems, Bio-Techne, Minneapolis, MN) was used. Protein was reconstituted at a concentration of 25 μg/ml in sterile PBS containing 0.1% BSA (Sigma-Aldrich) and added to PBMCs at various concentrations up to 500 ng/ml either alone or together with Z-AAT (0.5 mg/ml) for 18 hr. Afterward, cells were used for RNA isolation, flow cytometry or Western blot analysis. For Western blot, PBMCs were lysed in RIPA buffer (Sigma-Aldrich), supplemented with protease inhibitor cocktail (Sigma-Aldrich). For some Western blot experiments, we extracted detergent resistant lipid raft associated proteins from insoluble cell fractions using UltraRIPA kit according to the supplier's instructions (BioDynamics Laboratory, Tokyo, Japan).

## Western blot

Equal amounts of lysed proteins were separated by 7.5% or 10% SDS-polyacrylamide gels (under reducing conditions for CX3CR1 and non-reducing for total AAT or AAT polymer analysis) prior to transfer onto polyvinylidene difluoride membranes (Merck-Millipore, Burlington, MA). Membranes were blocked for 1 hr with 5% low fat milk (Carl Roth, Karlsruhe, Germany) followed by overnight incubation at 4°C with specific primary antibodies: polyclonal rabbit anti-human AAT (1:800) (DAKO A/S, Glostrup, Denmark), mouse monoclonal anti-AAT polymer antibody (clone 2C1, 1:500, Hycult Biotech, Uden, The Netherlands), rabbit polyclonal anti-CX3CR1 (1:500, Abcam, Cambridge, UK), or HRP-conjugated monoclonal anti-β-actin antibody (1:20,000, Sigma-Aldrich, Merck, St. Louis, MO) for a loading control. The immune complexes were visualized with anti-rabbit or anti-mouse HPR-conjugated secondary antibodies (DAKO A/S) and enhanced by Clarity Western ECL Substrate (Bio-Rad, Hercules, CA). Images were acquired by using Chemidoc Touch imaging system (BioRad) under optimal exposure conditions and processed using Image Lab v5.2.1 software (BioRad). For quantification, the signal intensity of the CX3CR1 protein band in each lane was divided by the corresponding β-actin band intensity (normalization factor or loading control). Afterward, the normalized signal of each lane was divided by the normalized target signal observed in the control sample to get the abundance of the CX3CR1 protein as a fold change relative to the control.

## ELISA

Plasma samples from 22 ZZ AATD and 21 non-AATD controls were analyzed for CX3CL1/Fractalkine using Duoset kit (R&D Systems, Minneapolis, MN, assay sensitivity 0.072 ng/ml, detection range 0.2–10 ng/ml). Cell-free culture supernatants were analyzed directly or stored at −80°C. ELISA Duoset kits for TNF-α (assay detection range 15.6–1000 pg/ml), IL-1β/IL-1F2 (assay detection range 3.91–250 pg/ml), and IL-6 (assay detection range 9.38–600 pg/ml) were purchased from R&D Systems (Minneapolis, MN) and were used according to the manufacturer's instructions. Plates were

measured on Infinite M200 microplate reader (Tecan, Männedorf, Switzerland). Measurements were carried out in duplicates.

## Flow cytometry analysis

PBMCs ($2 \times 10^6$ cells per condition) were incubated with LPS (1 µg/ml), M-AAT (1 mg/ml), or Z-AAT (0.5 mg/ml) for 18 hr. Staining was performed with phycoerythrin (PE)-conjugated mouse monoclonal anti-CX3CR1 antibody (clone 2A9-1 Invitrogen, Thermo Fisher Scientific, Carlsbad, CA), fluorescein (FITC)-conjugated mouse monoclonal anti-CD14 antibody (clone TuK4, Life Technologies, Thermo Fisher Scientific, Carlsbad, CA), allophycocyanin (APC)-conjugated mouse monoclonal anti-CD16 antibody (clone 3G8, Immunotools, Friesoythe, Germany), or BV-480-conjugated anti-CD56 mouse monoclonal antibody (Clone NCAM16.2, BD Biosciences, San Jose, CA) alone or in combinations. Dead cells were excluded by a staining with 7-amino-actinomycin D. Samples were measured on a BD FACSAria Fusion machine and analyzed with FlowJo v10 (Becton, Dickinson and Company, Franklin Lakes, NJ). The gating strategy is shown in *Figure 3—figure supplement 1*.

## Immunofluorescence confocal laser microscopy

Human total PBMCs ($2 \times 10^6$) were plated onto glass coverslips and incubated alone or with Z-AAT polymers (0.5 mg/ml) in RPMI medium for 18 hr at 37°C and 5% $CO_2$. Cells were then washed with PBS, fixed with 3% paraformaldehyde in PBS for 20 min, and continued with or without permeabilization with 0.5% Triton X-100 in PBS for 5 min at RT. For immunolabeling, cells were co-incubated with primary antibodies against human CX3CR1 (rabbit polyclonal IgG [1:500], Abcam, Cambridge, UK) and anti-AAT polymer antibody, LG96 (1:5000, mouse monoclonal) for 1 hr, at RT. After washing, the cells were incubated with corresponding secondary antibodies (1:1000) conjugated to Alexa-Fluor-488 (goat anti-rabbit) or AlexaFlour-594 (goat anti-mouse) both from Thermo Fisher Scientific, Rockford, IL. After final wash, the cells were mounted on microscope slides using ProLong Gold Antifade Mountant with DAPI (Thermo Fisher Scientific, Carlsbad, CA). Images were acquired using confocal laser microscope FluoView 1000 (Olympus, Shinjuku, Japan) equipped with a 60× oil immersion objective and differential interference contrast in sequential mode. Confocal z-stacks were collected with a 0.25 µm increment.

## Statistics

Data were analyzed and visualized by using Sigma Plot 14.0. One-tailed Student's t-test was applied to compare two sample means on one variable. When more than two groups were involved in the comparison, one-way ANOVA was used. Data were presented as mean (SD). If normality test failed, the nonparametric Kruskal-Wallis one-way analysis or Mann-Whitney rank sum test was performed, and data were presented as median (range). For correlation analysis, the Pearson's linear correlation method was used to measure the correlation for a given pair. A p-value of less than 0.05 was considered significant.

## Acknowledgements

We thank the German society Alpha1 Deutschand eV and society members for support.

## Additional information

### Competing interests

Tobias Welte: reports grants from German Ministry of Research and Education, during the conduct of the study; personal fees from Grifols, CSL Behring, outside the submitted work. Pavel Strnad: reports grants and personal fees from CSL Behring, grants and personal fees from Grifols Inc, personal fees from Dicerna Inc, grants from Vertex, grants from Arrowhead, outside the submitted work. Jan Stolk: reports grants from CSL Behring, grants from Kamada LtD, during the conduct of the study. The other authors declare that no competing interests exist.

## Funding

| Funder | Grant reference number | Author |
| --- | --- | --- |
| Deutsche Forschungsgemeinschaft | STR 1095/6-1 | Pavel Strnad |
| Deutsche Zentrum für Lungenforschung | 82DZL002A | Sabina Janciauskiene |
| Deutsche Forschungsgemeinschaft | SFB/TRR57 | Pavel Strnad |

The funders had no role in study design, data collection and interpretation, or the decision to submit the work for publication.

## Author contributions

Srinu Tumpara, Data curation, Validation, Investigation, Methodology, Writing - original draft, Writing - review and editing; Matthias Ballmaier, Elena Korenbaum, Data curation, Investigation, Visualization, Methodology, Writing - review and editing; Sabine Wrenger, Data curation, Investigation, Visualization, Methodology, Writing - original draft, Writing - review and editing; Mandy König, Matthias Lehmann, Ralf Lichtinghagen, Beatriz Martinez-Delgado, David DeLuca, Data curation, Investigation, Methodology, Writing - review and editing; Nils Jedicke, Malin Fromme, Data curation, Investigation, Writing - review and editing; Tobias Welte, Conceptualization, Resources, Investigation, Writing - review and editing; Pavel Strnad, Conceptualization, Resources, Data curation, Funding acquisition, Investigation, Writing - review and editing; Jan Stolk, Conceptualization, Resources, Data curation, Supervision, Validation, Investigation, Writing - original draft, Project administration, Writing - review and editing; Sabina Janciauskiene, Conceptualization, Resources, Data curation, Supervision, Funding acquisition, Validation, Investigation, Visualization, Writing - original draft, Project administration, Writing - review and editing

## Author ORCIDs

Srinu Tumpara (iD) https://orcid.org/0000-0003-0363-3914
Matthias Ballmaier (iD) http://orcid.org/0000-0002-1352-5995
Sabine Wrenger (iD) https://orcid.org/0000-0002-9733-6162
Beatriz Martinez-Delgado (iD) http://orcid.org/0000-0001-6834-350X
David DeLuca (iD) http://orcid.org/0000-0002-0141-9116
Sabina Janciauskiene (iD) https://orcid.org/0000-0001-7687-5258

## Ethics

Human subjects: The institutional review board of Aachen University (EK 173/15) provided ethical approval for individuals recruited in Germany. Leiden University Medical Center provided ethical approval (project P00.083 and P01.101) for the second study group. For all individuals detailed medical records data were anonymized. All participants issued a written informed consent according to the ethical guidelines of the Helsinki Declaration (Hong Kong Amendment) as well as Good Clinical Practice (European guidelines).

## Decision letter and Author response

Decision letter https://doi.org/10.7554/eLife.64881.sa1
Author response https://doi.org/10.7554/eLife.64881.sa2

# Additional files

## Supplementary files

• Transparent reporting form

### Data availability

All data generated or analysed during this study are included in the manuscript and supporting files. Source data files have been provided for Figures.

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
