## [Decision Letter]

Thank you for submitting your article "Polymerization of misfolded protein lowers CX3CR1 expression in human PBMCs" for consideration by *eLife*. Your article has been reviewed by 2 peer reviewers, one of whom is a member of our Board of Reviewing Editors, and the evaluation has been overseen Satyajit Rath as the Senior Editor. The reviewers have opted to remain anonymous.

The reviewers have discussed the reviews with one another, and the Reviewing Editor has drafted this decision to help you prepare a revised submission.

The reviewers agree that the implications of your work are broad and carry clinical significance, although there are major concerns over the lack of sufficient molecular level understanding of the regulation of CX3CR1's cell surface expression by Z-AAT polymers, for consideration for publication in *eLife*. Considering these concerns of the reviewers, we can only consider the manuscript further after a major revision if you sufficiently address all the concerns as outlined below.

Summary:

The manuscript by Tumpara S. et al. shows that polymers of mutated α-1 antitrypsin (AAT) affect expression and internalization of CX3CR1, presenting a novel example of CX3CR1 regulation by misfolded proteins and what could be a novel immunomodulatory function of protein α-1 antitrypsin. Using human PBMCs and plasma purified AAT in its wild type monomeric and mutated polymeric forms, authors show that Z AAT polymers reduce CX3CR1 expression and its surface expression. As CX3CR1 has been previously found to be regulated by Tau and amyloid-β, this finding poses a question if other conformationally altered extracellular proteins could have the same property. It also opens a question how this process is affected by inflammatory conditions and what are its implications for individuals with Z AAT. In addition to the novel mechanism of CX3CR1 regulation, these findings could be of relevance for better understanding of molecular pathophysiology associated with AAT deficiency (AATD) adding a potentially important step to the mechanism of AATD pathology. Furthermore, these findings could also have relevance in other conditions in which CX3CR1 axis and AAT have been implicated (e.g. cancer).

Essential revisions:

1. The protein level of CX3XR1 in total cell lysate is high although the gene expression is significantly low in AATD patients or Z-AAT treated cells which has been explained by showing Z-AAT mediated CX3XR1's association with lipid rafts. How specific is this effect? Would any oligomeric protein which can bind to CX3CR1, have similar effect? What is the effect with treatment with monomeric Z-AAT on CX3CR1 gene expression and protein level?

2. There is no explanation provided for decreased expression CX3CR1 in AATD patients or after in vitro polymeric Z-AAT treatment except for indirect connection with increased CD14 expression (CD14++ cells are known to decrease CX3CR1 expression). What is the possible mechanism of such drastic decrease in expression level of CX3CR1?

3. Are there any changes in CX3CR1 expression after fractaline challenge of the PBMCs treated with polymeric Z-AAT?

4. Is there any change in CX3CR1 expression and cell surface expression in cells having intracellular Z-AAT polymers?

[Editors' note: further revisions were suggested prior to acceptance, as described below.]

Thank you for resubmitting your work entitled "Polymerization of misfolded Z α-1antitrypin protein lowers CX3CR1 expression in human PBMCs" for further consideration by *eLife*. Your revised article has been reviewed by two reviewers, one of whom is a member of our Board of Reviewing Editors, and the evaluation has been overseen by Satyajit Rath as the Senior Editor.

Although both the reviewers have agreed that the manuscript has been improved by adding further experimental evidence for reduced cell surface expression of CX3CR1 by ZZ-AAT polymers and additional experiments, there are some issues, as outlined below, which need to be addressed before a final decision can be made.

1. The explanation for significant decrease in mRNA expression levels of CX3CR1 in patient PBMCs by ZZ-AAT polymers is not properly supported experimentally. At the very least, an acknowledgement and a plausible explanation of such an important observation is essential.

2. A discussion (supported by experiments if possible) on the fate of internalized CX3CR1 upon removal of ZZ-AAT polymers from surrounding media should be included to increase the impact of the study.

3. The submission is being considered as a short report. It is critical to maintain the appropriate format (such as limiting the total number of main display items within )5, in the revision.

The detailed recommendations from both reviewers are being provided below for ease of response.

*Reviewer #1:*

In the revised version of manuscript 13-11-2020-RA-*eLife*-64881R1, Tumpara et al. has successfully addressed the observation of decreased cell surface expression of CX3CR1 on PBMCs in patients of α-1 antitrypsin deficiency (AATD) with circulating polymeric forms of AAT (Z-AAT) protein. They have shown the exclusive role of polymeric ZZ-AAT in CX3CR1 internalization in comparison to monomeric form of the protein or the CX3CL1. The authors mentioned that this finding is already known for other aggregated proteins like Tau, so this finding is not sufficiently novel. Importantly, the most intriguing finding of significant decrease in mRNA level expression of CX3CR1 is not explained except for an indirect correlation with increased CD14 expression by the ZZ-AATD treated cells. This finding needs further explanation. Authors have only explained about reduction of cell surface expression of CX3CR1 which is not satisfactory to explain the decreased mRNA level expression of CX3CR1 with polymeric ZZ-AATD exclusively and its implication in patients of AATD. To be acceptable for publication, this important observation needs to be explained and its implication in patients of AATD needs to be discussed in detail.

*Reviewer #3:*

Manuscript provides data showing that Z AAT induces reduced CX3CR1 gene expression and protein surface expression while increasing intracellular levels of the protein and new data provided by the authors show that the effect on CX3CR1 expression is executed specifically by Z AAT polymer and not by its monomeric form or polymeric variant of WT protein. Experiments including CXCL1 were added and new colocalization experiment provides an explanation for increased amount of CX3CR1 in cell lysates treated with Z AAT and I believe the manuscript is suitable for publication. It would be interesting to know if upon removal of ZAAT from surrounding media CX3CR1 could be recycled back to the surface or is the interaction more permanent (ZAAT induces increase in intracellular CXCR1 compared to CX3CL1), however that would be beyond the scope of this manuscript. Given the limit of 5 figures for short report the authors can consider moving Figure 5 showing gene expression in presence of CX3CL1, as well as Key Resources Table to supplement or possibly merging Figure 4 with Figure 2.

---

## [Author Response]

Essential revisions:1. The protein level of CX3XR1 in total cell lysate is high although the gene expression is significantly low in AATD patients or Z-AAT treated cells which has been explained by showing Z-AAT mediated CX3XR1's association with lipid rafts. How specific is this effect? Would any oligomeric protein which can bind to CX3CR1, have similar effect? What is the effect with treatment with monomeric Z-AAT on CX3CR1 gene expression and protein level?

Thank you very much for this important point. Other pathogenic oligomeric proteins, which bind to CX3CR1, may have a similar effect as Z-AAT polymers. For example, it has been shown that CX3CR1 in microglia acts as surface receptor for extracellular Tau oligomers and promotes its internalization (Bolós M, Llorens-Martín M, Perea JR, Jurado-Arjona J, Rábano A, Hernández F, et al. Absence of CX3CR1 impairs the internalization of Tau by microglia. Mol Neurodegener. 2017;12(1):59). However, it is difficult to predict if this phenomenon applies to all misfolded proteins.

As suggested, we also tested Z-AAT monomers in our PBMCs model. However, monomer showed no effect on CX3CR1 mRNA and protein levels. We included these data into the manuscript. We also tested artificially generated M-AAT polymers by heating M-AAT for 3h at 60°C. This method is well-known and published many times as useful to induce polymers of normal M-AAT. However, M-AAT polymers showed no effect on CX3CR1 levels. This shows that only certain polymers, probably dependent on their size and/or structural properties can interact with CX3CR1 and regulate its surface expression and intracellular levels. However, the mechanisms behind this regulation requires detailed studies in vivo. We added new data regarding effects of Z-AAT monomer and M-AAT heat-induced polymer on CX3CR1 receptor levels in PBMCs, and included additional discussion and references into the manuscript.

2. There is no explanation provided for decreased expression CX3CR1 in AATD patients or after in vitro polymeric Z-AAT treatment except for indirect connection with increased CD14 expression (CD14++ cells are known to decrease CX3CR1 expression). What is the possible mechanism of such drastic decrease in expression level of CX3CR1?

Thank you for this point. We included additional data and putative explanations regarding reduced CX3CR1 expression in ZZ-AAT PBMCs. Sustained CX3CR1 down-regulation in ZZ-AATD individuals seems to result from the circulating Z-AAT polymers which interact with cells and get internalized. Similar like in case of Tau polymers, Z-AAT polymers seems to internalize together with CX3CR1 resulting in reduced cell surface expression of CX3CR1. We performed additional experiments using laser scanning confocal microscopy and demonstrate that in many cells Z-AAT polymers localize with CX3CR1. A new figure and additional data are included into the manuscript.

We also included a statement into the manuscript that, although both Z-AAT polymers and LPS induce CD14 expression and similarly affect CX3CR1 expression and protein levels, our results imply that Z-AAT polymers and LPS do not share the same signalling mechanisms. LPS induces cell activation as measured by induced cytokine production whereas under the same experimental conditions Z-AAT polymers did not induce cytokines.

3. Are there any changes in CX3CR1 expression after fractaline challenge of the PBMCs treated with polymeric Z-AAT?

Thank you for this question. We performed additional experiments, in which total PBMCs were pre-incubated first with various concentrations of soluble CX3CL1 followed by incubation without or with constant amount of Z-AAT polymers for 18 hours. According to our results, soluble CX3CL1 alone was unable to affect the expression of its own receptor and had no effect on Z-AAT polymers. Independent of the presence of soluble CX3CL1 (even at very high concentration such as 500 ng/ml) Z-AAT polymers down-regulated CX3CR1 mRNA expression and induce intracellular levels of CX3CR1 protein. Our finding that soluble CX3CL1 has no effect on its receptor is in concordance with data published by Pachot et al. J Immunol, 2008, 180 (9) 6421-6429. However, there are reports showing that soluble CX3CL1 induces an internalization of CX3CR1 (Arterioscler Thromb Vasc Biol. 2014, 34(12): 2554–2562). Data from our new experiments and additional discussion are included into the manuscript.

4. Is there any change in CX3CR1 expression and cell surface expression in cells having intracellular Z-AAT polymers?

Thank you for the question. Our findings are based on cells isolated from ZZ-AAT deficiency individuals who naturally contain Z-polymers in all cells producing AAT as well as extracellularly, in all biological fluids. Monocytes and macrophages express AAT, therefore our PBMCS from ZZ AAT deficiency individuals include cells having Z-AAT polymers and low CX3CR1 expression. We now show that circulating ZZ polymers, which are markers for Z- AAT deficiency, can be taken up by the cells. We performed confocal laser scanning microscopy and confirmed that Z-AAT polymer enters cells and in part localize together with CX3CR1 receptor. We included these data into the manuscript. Therefore, even if polymers are intracellularly not produced their can be taken up from the circulation. Our current experiments show that cells can take up Z-AAT and lower CX3CR1 expression. Hence, theoretically cells having intracellular Z-AAT polymers will have lower surface expression of CX3CR1.

Unfortunately, due to limited access to rare ZZ patients, especially during the covid-19 times, we are not able to obtain new blood for monocyte or macrophage isolation and polymer analysis. We also do not know how cell isolation and sorting would affect intracellular polymers of AAT. A detailed comparison of cells with and without Z-AAT polymers regarding CX3CR1 expression would be of interest to perform in the further.